# Transformer-XL: Language Modeling with Longer-Term Dependency

## Abstract

We propose a novel neural architecture, *Transformer-XL*, for modeling longer-term dependency. To address the limitation of fixed-length contexts, we introduce a notion of recurrence by reusing the representations from the history. Empirically, we show state-of-the-art (SoTA) results on both word-level and character-level language modeling datasets, including WikiText-103, One Billion Word, Penn Treebank, and enwiki8. Notably, we improve the SoTA results from 1.06 to 0.99 in bpc on enwiki8, from 33.0 to 18.9 in perplexity on WikiText-103, and from 28.0 to 23.5 in perplexity on One Billion Word. Performance improves when the attention length increases during evaluation, and our best model attends to up to 1,600 words and 3,800 characters. To quantify the effective length of dependency, we devise a new metric and show that on WikiText-103 Transformer-XL manages to model dependency that is about 80% longer than recurrent networks and 450% longer than Transformer. Moreover, Transformer-XL is up to 1,800+ times faster than vanilla Transformer during evaluation.

## 1 Introduction

Language modeling is among the important problems that require modeling long-term dependency. However, it has been a challenge to equip neural networks with the capability to model long-term dependency in sequential data. Recurrent neural networks (RNNs), in particular Long Short-Term Memory (LSTM) networks (Hochreiter & Schmidhuber, 1997), have been a standard solution for language modeling and obtained strong results on multiple benchmarks. However, RNNs are difficult to optimize due to gradient vanishing and explosion (Hochreiter et al., 2001), and the introduction of gating in LSTMs and the gradient clipping technique (Graves, 2013; Pascanu et al., 2012) might not be sufficient to fully address this issue. It was shown that LSTM language models use 200 context words on average (Khandelwal et al., 2018), indicating room for further improvement.

On the other hand, the direct connections between long-distance word pairs baked in attention mechanisms might ease optimization and enable the learning of long-term dependency (Bahdanau et al., 2014; Vaswani et al., 2017). Recently, Al-Rfou et al. (2018) designed a set of auxiliary losses to train deep Transformer networks for character-level language modeling, which outperform LSTMs by a large margin. However, the context length is fixed to hundreds of characters and thus it is not possible to model longer-term dependency. Moreover, it is not clear how the model performs on word-level language modeling data, as the granularity changes.

To address the limitation of fixed-length contexts, we propose a new architecture called Transformer-XL (meaning extra long). We introduce the notion of recurrence into our deep self-attention network. In particular, instead of computing the hidden states from scratch for each new segment, we reuse the hidden states obtained in previous segments. The reused hidden states serve as memory for the current segment, which builds up a recurrent connection between the segments. As a result, modeling very long-term dependency becomes possible because information can be propagated through the recurrent connections. Importantly, to reuse the hidden states, it is necessary to use relative positional encodings rather than absolute ones. As an additional technical contribution, we introduce a simple and effective relative positional encoding formulation that generalizes to attention lengths longer than the one observed during training.

Transformer-XL obtained strong results on four datasets, varying from word-level to character-level language modeling. Transformer-XL improves the previous state-of-the-art (SoTA) results from

33.0 to 18.9 in perplexity on WikiText-103, from 1.06 to 0.99 in bpc on enwiki8, and from 28.0 to 23.5 in perplexity on One Billion Word. On small data, Transformer-XL also achieves a perplexity of 54.5 on Penn Treebank, which is SoTA when comparable settings are considered.

We use two methods to quantitatively study the effective lengths of Transformer-XL and the baselines. Similar to Khandelwal et al. (2018), we gradually increase the attention length at test time until no further noticeable improvement ($\sim$0.1% relative gains) can be observed. Our best model in this settings use attention lengths of 1,600 and 3,800 on WikiText-103 and enwiki8 respectively. In addition, since the effective context length of Transformer-XL can be longer than the attention length due to our recurrent formulation, we devise a metric called *Relative Effective Context Length* (RECL) that aims to perform a fair comparison of the gains brought by increasing the context lengths for different models. In this setting, Transformer-XL learns a RECL of 900 words on WikiText-103, while the numbers for recurrent networks and Transformer are only 500 and 128.

## 2   RELATED WORK

In the last few years, the field of language modeling has witnessed many significant advances, including but not limited to devising novel architectures to better encode the context (Bengio et al., 2003; Mikolov et al., 2010; Zilly et al., 2016; Krause et al., 2016; Grave et al., 2016b; Dauphin et al., 2016; Chung et al., 2016; Merity et al., 2016; Kalchbrenner et al., 2016; Al-Rfou et al., 2018), improving regularization and optimization algorithms Zaremba et al. (2014); Inan et al. (2016); Press & Wolf (2016); Merity et al. (2017); Gal & Ghahramani (2016), speeding up the Softmax computation (Morin & Bengio, 2005; Kuchaiev & Ginsburg, 2017; Grave et al., 2016a; Jozefowicz et al., 2016), and enriching the output distribution family (Yang et al., 2017; Kanai et al., 2018).

To capture the long-range context in language modeling, a line of work directly feeds a representation of the wider context into the network as an additional input. Existing works range from ones where context representations are manually defined (Mikolov & Zweig, 2012; Ji et al., 2015; Wang & Cho, 2015) to others that rely on document-level topics learned from data (Dieng et al., 2016; Wang et al., 2017).

More broadly, in generic sequence modeling, how to capture long-term dependency has been a long-standing research problem. From this perspective, since the ubiquitous adaption of LSTM, many efforts have been spent on relieving the vanishing gradient problem, including better initialization (Le et al., 2015), additional loss signal (Trinh et al., 2018) and others that modify the internal architecture of RNNs to ease the optimization Mikolov et al. (2014); Koutnik et al. (2014); Wu et al. (2016); Li et al. (2018). Different from them, our work is based on the Transformer architecture and shows that language modeling as a real-world task benefits from the ability to learn longer-term dependency.

## 3   MODEL

Given a corpus of tokens $\mathbf{x} = (x_1, \ldots, x_T)$, the task of language modeling is to estimate the joint probability $P(\mathbf{x})$, which is often auto-regressively factorized as $P(\mathbf{x}) = \prod_t P(x_t \mid \mathbf{x}_{<t})$. With the factorization, the problem reduces to estimating each conditional factor. In this work, we stick to the standard neural approach to modeling the conditional probability. Specifically, a trainable neural network is used to encode the context $\mathbf{x}_{<t}$ into a fixed size hidden state, which is multiplied with the word embeddings to obtain the logits. The logits are then fed into the Softmax function, yielding a categorical probability distribution over the next token.

### 3.1   VANILLA TRANSFORMER LANGUAGE MODELS

In order to apply Transformer or self-attention to language modeling, the central problem is how to train a Transformer to effectively encode an arbitrarily long context into a fixed size representation. Given infinite memory and computation, a simple solution would be to process the entire context sequence using an unconditional Transformer decoder, similar to a feed-forward neural network. However, this is usually infeasible with the limited resource in practice.

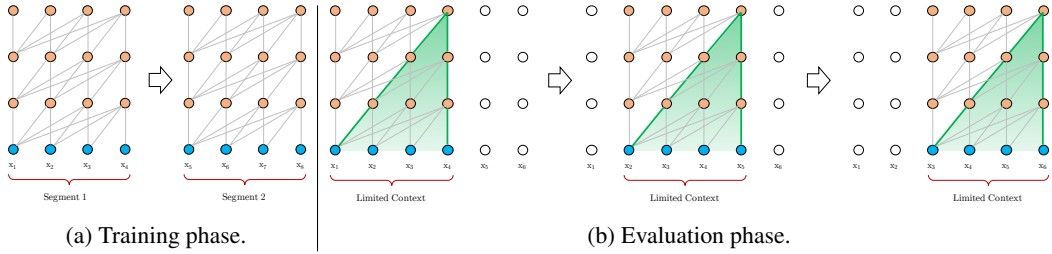

(a) Training phase.

(b) Evaluation phase.

Figure 1: Illustration of the vanilla model with a segment length 4.

One feasible but crude approximation is to split the entire corpus into shorter segments of manageable sizes, and only train the model within each segment, ignoring all contextual information from previous segments. This is the idea adopted by Al-Rfou et al. (2018). We call it the *vanilla model* and visualize it in Fig. 1a. Under this training paradigm, information never flows across segments in either the forward or backward pass. Hence, the largest possible dependency length is upper bounded by the segment length, which is a few hundred on character-level language modeling (Al-Rfou et al., 2018). Therefore, although the self-attention mechanism is less affected by the vanishing gradient problem compared to RNNs, the vanilla model is not able to fully exploit this optimization advantage.

During evaluation, at each step, the vanilla model also consumes a segment of the same length as in training, but only makes one prediction at the last position. Then, at the next step, the segment is shifted to the right by only one position, and the new segment has to be processed all from scratch. As shown in Fig. 1b, this procedure ensures that each prediction utilizes the longest possible context exposed during training.

## 3.2 SEGMENT-LEVEL RECURRENCE WITH STATE REUSE

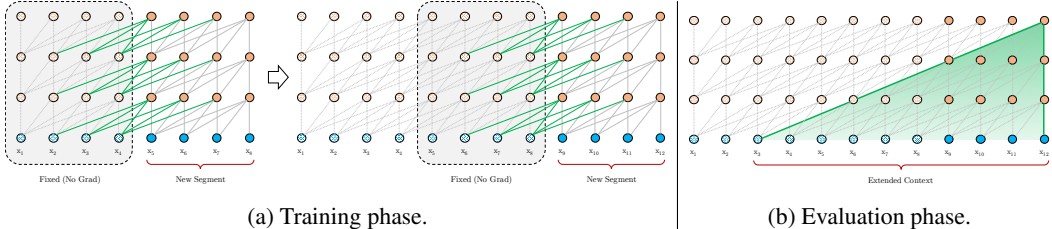

(a) Training phase.

(b) Evaluation phase.

Figure 2: Illustration of the Transformer-XL model with a segment length 4.

To address the limitation of using fixed-length contexts, we propose to introduce a recurrence mechanism to the Transformer architecture. During training, the hidden state sequence computed for the previous segment is *fixed* and *cached* to be reused as an extended context when the model processes the next new segment, as shown in Fig. 2a. Although the gradient still remains within a segment, this additional input allows the network to exploit information in the history, leading to an ability of modeling longer-term dependency. Formally, let the two consecutive segments of length $L$ be $\mathbf{s}_\tau = [x_{\tau,1}, \cdots, x_{\tau,L}]$ and $\mathbf{s}_{\tau+1} = [x_{\tau+1,1}, \cdots, x_{\tau+1,L}]$ respectively. Denoting the $n$-th layer hidden state sequence produced for the $\tau$-th segment $\mathbf{s}_\tau$ by $\mathbf{h}_\tau^n \in \mathbb{R}^{L \times d}$, where $d$ is the hidden dimension. Then, the $n$-th layer hidden state for segment $\mathbf{s}_{\tau+1}$ is produced as follows,

$$\widetilde{\mathbf{h}}_{\tau+1}^{n-1} = [\text{SG}(\mathbf{h}_\tau^{n-1}) \circ \mathbf{h}_{\tau+1}^{n-1}], \qquad \text{(extended context)}$$

$$\mathbf{q}_{\tau+1}^n, \mathbf{k}_{\tau+1}^n, \mathbf{v}_{\tau+1}^n = \mathbf{h}_{\tau+1}^{n-1}\mathbf{W}_q^\top, \widetilde{\mathbf{h}}_{\tau+1}^{n-1}\mathbf{W}_k^\top, \widetilde{\mathbf{h}}_{\tau+1}^{n-1}\mathbf{W}_v^\top, \qquad \text{(query, key, value vectors)}$$

$$\mathbf{h}_{\tau+1}^n = \text{Transformer-Layer}\left(\mathbf{q}_{\tau+1}^n, \mathbf{k}_{\tau+1}^n, \mathbf{v}_{\tau+1}^n\right). \qquad \text{(self-attention + feed-forward)}$$

where the function $\text{SG}(\cdot)$ stands for stop-gradient, the notation $[\mathbf{h}_u \circ \mathbf{h}_v]$ indicates the concatenation of two hidden sequences along the length dimension, and $\mathbf{W}_.$ denotes model parameters. Compared to the standard Transformer, the critical difference lies in that the key $\mathbf{k}_{\tau+1}^n$ and value $\mathbf{v}_{\tau+1}^n$ are

conditioned on the extended context $\widetilde{\mathbf{h}}_{\tau+1}^{n-1}$ and hence $\mathbf{h}_\tau^{n-1}$ cached from the previous segment. We emphasize this particular design by the green paths in Fig. 2a.

With this recurrence mechanism applied to every two consecutive segments of a corpus, it essentially creates a segment-level recurrence in the hidden states. As a result, the effective context being utilized can go way beyond just two segments. However, notice that the recurrent dependency between $\mathbf{h}_{\tau+1}^n$ and $\mathbf{h}_\tau^{n-1}$ shifts one layer downwards per-segment, which differs from the same-layer recurrence in conventional RNN-LM. Consequently, the largest possible dependency length grows linearly w.r.t. the number of layers as well as the segment length, as visualized by the shaded area in Fig. 2b. This is analogous to truncated BPTT (Mikolov et al., 2010), a technique developed for training RNN-LM. However, different from truncated BPTT, our method caches a sequence of hidden states instead of the last one, and should be applied together with the relative positional encoding technique described in Section 3.3.

Besides the extra long context, another benefit that comes with the recurrence scheme is significantly faster evaluation. Specifically, during evaluation, the representations from the previous segments can be reused instead of being computed from scratch as in the case of the vanilla model. In our experiments on enwiki8, Transformer-XL is up to 1,800+ times faster than the vanilla model during evaluation (see Appendix 4.4 for more comparisons).

Finally, notice that the recurrence scheme does not need to be restricted to only the previous segment. In theory, we can cache as many previous segments as the GPU memory allows, and reuse all of them as the extra context when processing the current segment. In practice, we cache a predefined length-$M$ old hidden states spanning (possibly) multiple segments, and refer to them as the memory $\mathbf{m}_\tau^n \in \mathbb{R}^{M \times d}$, due to a clear connection to the memory augmented neural networks (Graves et al., 2014; Weston et al., 2014). In our experiments, we set $M$ equal to the segment length during training, and increase it by multiple times during evaluation.

## 3.3 RELATIVE POSITIONAL ENCODINGS

While we found the idea presented in the previous subsection very appealing, there is a crucial technical challenge we haven't solved in order to reuse the hidden states. That is, how can we keep the positional information coherent when we reuse the states? Recall that, in the standard Transformer, the information of sequence order is provided by a set of positional encodings, denoted as $\mathbf{U} \in \mathbb{R}^{L_{\max} \times d}$, where the $i$-th row $\mathbf{U}_i$ corresponds to the $i$-th *absolute* position within a segment and $L_{\max}$ prescribes the maximum possible length to be modeled. Then, the actual input to the Transformer is the element-wise addition of the word embeddings and the positional encodings. If we simply adapt this positional encoding to our recurrence mechanism introduced above, the hidden state sequence would be computed schematically by

$$\mathbf{h}_{\tau+1} = f(\mathbf{h}_\tau, \mathbf{E}_{\mathbf{s}_{\tau+1}} + \mathbf{U}_{1:L}) \quad \text{and} \quad \mathbf{h}_\tau = f(\mathbf{h}_{\tau-1}, \mathbf{E}_{\mathbf{s}_\tau} + \mathbf{U}_{1:L}),$$

where $\mathbf{E}_{\mathbf{s}_\tau} \in \mathbb{R}^{L \times d}$ is the word embedding sequence of $\mathbf{s}_\tau$, and $f$ represents a transformation function. Notice that, both $\mathbf{E}_{\mathbf{s}_\tau}$ and $\mathbf{E}_{\mathbf{s}_{\tau+1}}$ are associated with the same positional encoding $\mathbf{U}_{1:L}$. As a result, the model has no information to distinguish the positional difference between $x_{\tau,j}$ and $x_{\tau+1,j}$ for any $j = 1, \ldots, L$, resulting in a sheer performance loss.

In order to avoid this failure mode, the fundamental idea is to only encode the *relative* positional information in the hidden states. Intuitively, when a query vector $q_{\tau,i}$ attends on the key vectors $\mathbf{k}_{\tau,\leq i}$ from the $i$-th position, it does not need to know the absolute position of each key vector to identify the order of the segment. Instead, it suffices to know the relative distance between each key vector $k_{\tau,j}$ and itself $q_{\tau,i}$, i.e. $i - j$.[1] Practically, one can create a set of relative positional encodings $\mathbf{R} \in \mathbb{R}^{L_{\max} \times d}$, where the $i$-th row $\mathbf{R}_i$ indicates a relative distance of $i$ between two positions. By injecting this distance information into the attention, the sequence order can be completely captured without any loss.

Previously, the idea of relative positional encodings has been explored in the context of machine translation (Shaw et al., 2018) and music generation (Huang et al., 2018). Here, we offer a different

---

[1] In this work, as we focus on language modeling, the attention is also causal, i.e., from right to left. Hence, the relative distance, defined as the right index minus the left index, is always non-negative. But it is straight-forward to extend this idea to the non-causal case.

derivation, arriving at a new form of relative positional encodings, which not only has a one-to-one correspondence to its absolute counterpart but also enjoys much better generalization empirically (see Section 4). Firstly, in the standard Transformer (Vaswani et al., 2017), the attention score between query $q_i$ and key vector $k_j$ within the same segment can be decomposed as

$$\mathbf{A}_{i,j}^{\mathrm{abs}} = q_i^\top k_j = \underbrace{\mathbf{E}_{x_i}^\top \mathbf{W}_q^\top \mathbf{W}_k \mathbf{E}_{x_j}}_{(a)} + \underbrace{\mathbf{E}_{x_i}^\top \mathbf{W}_q^\top \mathbf{W}_k \mathbf{U}_j}_{(b)} + \underbrace{\mathbf{U}_i^\top \mathbf{W}_q^\top \mathbf{W}_k \mathbf{E}_{x_j}}_{(c)} + \underbrace{\mathbf{U}_i^\top \mathbf{W}_q^\top \mathbf{W}_k \mathbf{U}_j}_{(d)}.$$

With the goal of removing all information about absolute positions, we propose to re-parameterize the four terms as follows

$$\mathbf{A}_{i,j}^{\mathrm{rel}} = \underbrace{\mathbf{E}_{x_i}^\top \mathbf{W}_q^\top \mathbf{W}_{k,E} \mathbf{E}_{x_j}}_{(a)} + \underbrace{\mathbf{E}_{x_i}^\top \mathbf{W}_q^\top \mathbf{W}_{k,R} \mathbf{R}_{i-j}}_{(b)} + \underbrace{u^\top \mathbf{W}_{k,E} \mathbf{E}_{x_j}}_{(c)} + \underbrace{v^\top \mathbf{W}_{k,R} \mathbf{R}_{i-j}}_{(d)}.$$

- The first change we make is to replace all appearances of the absolute positional embedding $\mathbf{U}_j$ for computing key vectors in term $(b)$ and $(d)$ with its relative counterpart $\mathbf{R}_{i-j}$. This essentially reflects the prior that only the relative distance matters for where to attend. Note that $\mathbf{R}$ is a sinusoid encoding matrix (Vaswani et al., 2017) without learnable parameters.

- Secondly, we introduce a trainable parameter $u \in \mathbb{R}^d$ to replace the query $\mathbf{U}_i^\top \mathbf{W}_q^\top$ in term $(c)$. In this case, since the query vector is the same for all query positions, it suggests that the attentive bias towards different words should remain the same regardless of the query position. With a similar reasoning, a trainable parameter $v \in \mathbb{R}^d$ is added to substitute $\mathbf{U}_i^\top \mathbf{W}_q^\top$ in term $(d)$.

- Finally, we deliberately separate the two weight matrices $\mathbf{W}_{k,E}$ and $\mathbf{W}_{k,R}$ for producing the content-based key vectors and location-based key vectors respectively.

Under the new parameterization, each term has an intuitive meaning: term $(a)$ represents content-based addressing, term $(b)$ captures a content-dependent positional bias, term $(c)$ governs a global content bias, and $(d)$ encodes a global positional bias.

In comparison, the formulation in Shaw et al. (2018) only has terms $(a)$ and $(b)$, dropping the two bias terms $(c)$ and $(d)$. Moreover, Shaw et al. (2018) merge the multiplication $\mathbf{W}_k \mathbf{R}$ into a single trainable matrix $\hat{\mathbf{R}}$, which abandons the inductive bias built into the original sinusoid positional encoding (Vaswani et al., 2017). In contrast, our relative positional embedding $\mathbf{R}$ adapts the sinusoid formulation. As a benefit of the inductive bias, a model trained on a memory of some certain length can automatically generalize to a memory several times longer during evaluation.

Equipping the recurrence mechanism with our proposed relative positional embedding, we finally arrive at the Transformer-XL architecture. For completeness, we summarize the computational procedure for a $N$-layer Transformer-XL with a single attention head below:

$$\text{For } n = 1, \ldots, N : \quad \widetilde{\mathbf{h}}_\tau^{n-1} = \left[ \mathrm{SG}(\mathbf{m}_\tau^{n-1}) \circ \mathbf{h}_\tau^{n-1} \right]$$

$$\mathbf{q}_\tau^n, \mathbf{k}_\tau^n, \mathbf{v}_\tau^n = \mathbf{h}_\tau^{n-1} {\mathbf{W}_q^n}^\top, \widetilde{\mathbf{h}}_\tau^{n-1} {\mathbf{W}_{k,E}^n}^\top, \widetilde{\mathbf{h}}_\tau^{n-1} {\mathbf{W}_v^n}^\top$$

$$\mathbf{A}_{\tau,i,j}^n = {\mathbf{q}_{\tau,i}^n}^\top \mathbf{k}_{\tau,j}^n + {\mathbf{q}_{\tau,i}^n}^\top \mathbf{W}_{k,R}^n \mathbf{R}_{i-j} + u^\top \mathbf{k}_{\tau,j} + v^\top \mathbf{W}_{k,R}^n \mathbf{R}_{i-j}$$

$$\mathbf{a}_\tau^n = \text{Masked-Softmax}(\mathbf{A}_\tau^n) \mathbf{v}_\tau^n$$

$$\mathbf{o}_\tau^n = \text{LayerNorm}(\mathbf{a}_\tau^n + \mathbf{h}_\tau^{n-1})$$

$$\mathbf{h}_\tau^n = \text{Positionwise-Feed-Forward}(\mathbf{o}_\tau^n)$$

with $\mathbf{h}_\tau^0 := \mathbf{E}_{\mathbf{s}_\tau}$ defined as the word embedding sequence. In addition, it is worth mentioning that a naive way to compute $\mathbf{A}$ requires computing $\mathbf{W}_{k,R}^n \mathbf{R}_{i-j}$ for all pairs $(i, j)$, whose cost is quadratic w.r.t. the sequence length. However, noticing that the value of $i - j$ only ranges from zero to the sequence length, we show a simple computation procedure in Appendix B, which reduces the cost to be linear w.r.t. the sequence length.

## 4 EXPERIMENTS

### 4.1 MAIN RESULTS

We apply Transformer-XL to a variety of datasets on both word-level and character-level language modeling to have a comparison with state-of-the-art systems, including WikiText-103 (Merity et al.,

| Model | #Params | Validation PPL | Test PPL |
|---|---|---|---|
| Grave et al. (2016b) – LSTM | - | - | 48.7 |
| Bai et al. (2018) – TCN | - | - | 45.2 |
| Dauphin et al. (2016) – GCNN-8 | - | - | 44.9 |
| Grave et al. (2016b) – LSTM + neural cache | - | - | 40.8 |
| Dauphin et al. (2016) – GCNN-14 | - | - | 37.2 |
| Merity et al. (2018) – 4-layer QRNN | 151M | 32.0 | 33.0 |
| Ours – Transformer-XL Standard | 151M | 23.1 | 24.0 |
| Ours – Transformer-XL Large | 257M | **18.2** | **18.9** |

Table 1: Comparison with state-of-the-art results on WikiText-103.

| Model | #Params | Test bpc |
|---|---|---|
| Ha et al. (2016) – LN HyperNetworks | 27M | 1.34 |
| Chung et al. (2016) – LN HM-LSTM | 35M | 1.32 |
| Zilly et al. (2016) – Recurrent highway networks | 46M | 1.27 |
| Mujika et al. (2017) – Large FS-LSTM-4 | 47M | 1.25 |
| Krause et al. (2016) – Large mLSTM | 46M | 1.24 |
| Knol (2017) – cmix v13 | - | 1.23 |
| Al-Rfou et al. (2018) – 12-layer Transformer | 44M | 1.11 |
| Ours – 12-layer Transformer-XL | 41M | **1.06** |
| Al-Rfou et al. (2018) – 64-layer Transformer | 235M | 1.06 |
| Ours – 18-layer Transformer-XL | 88M | 1.03 |
| Ours – 24-layer Transformer-XL | 277M | **0.99** |

Table 2: Comparison with state-of-the-art results on enwiki8.

2016), enwiki8 (LLC, 2009), One Billion Word (Chelba et al., 2013), and Penn Treebank (Mikolov & Zweig, 2012).

WikiText-103 is the largest available word-level language modeling benchmark with long-term dependency. It contains 103M training tokens from 28K articles, with an average length of 3.6K tokens per article, which allows testing the ability of long-term dependency modeling. We set the attention length to 384 during training and 1600 during evaluation. As shown in Table 1, Transformer-XL reduces the previous SoTA perplexity from 33.0 to 18.9, which demonstrates that our new architecture is substantially better than previous models on word-level datasets that require modeling long-term dependency.

The dataset enwiki8 contains 100M bytes of unprocessed Wikipedia text. We compare our architecture with the previous SoTA results in Table 2. The 12-layer Transformer-XL achieves a new SoTA result in the setting with a model size constraint, outperforming the 12-layer Transformer model from Al-Rfou et al. (2018) by 0.05. Both Transformer variants have a large margin over conventional RNN-based models. Notably, our 12-layer architecture achieves the same result as the 64-layer network from Al-Rfou et al. (2018), using only 17% of the parameter budget. In order to see whether better performance can be obtained by increasing the model size, we train 18-layer and 24-layer Transformer-XLs with increased model sizes. We set the attention length at 784 during training and 3,800 during evaluation. We obtained a new SoTA result and our method is the first to break through 1.0 on widely-studied character-level benchmarks. Different from Al-Rfou et al. (2018), Transformer-XL does not need any auxiliary losses, and thus all benefits are credited to a better architecture.

One Billion Word does not preserve any long-term dependency because sentences have been shuffled. Consequently, this dataset mainly tests the ability of modeling only short-term dependency. The comparison between Transformer-XL and the other methods is shown in Table 4. We improve the SoTA from 28.0 to 23.5 among single-model results, with much less parameters.

We also report the results on word-level Penn Treebank in Table 3. Similar to AWD-LSTM (Merity et al., 2017), we apply variational dropout and weight average to Transformer-XL. With proper regu-

| Model | #Params | Dev PPL | Test PPL |
|---|---|---|---|
| Inan et al. (2016) – Tied Variational LSTM + augmented loss | 24M | 75.7 | 73.2 |
| Zilly et al. (2016) – Variational RHN | 23M | 67.9 | 65.4 |
| Zoph & Le (2016) – NAS Cell | 25M | - | 64.0 |
| Merity et al. (2017) – AWD-LSTM | 24M | 60.7 | 58.8 |
| Pham et al. (2018) – Efficient NAS | 24M | 60.8 | 58.6 |
| Liu et al. (2018) – Differentiable NAS | 23M | 58.3 | 56.1 |
| Yang et al. (2017) – AWD-LSTM-MoS | 22M | 58.08 | 55.97 |
| Melis et al. (2018) – 2-layer skip-LSTM + dropout tuning | 24M | 57.1 | 55.3 |
| Ours – Transformer-XL | 24M | **56.72** | **54.52** |
| Merity et al. (2017) – AWD-LSTM + finetuning[†] | 24M | 60.0 | 57.3 |
| Yang et al. (2017) – AWD-LSTM-MoS + finetuning[†] | 22M | **56.54** | **54.44** |

Table 3: Comparison with state-of-the-art results on Penn Treebank word-level language modeling. † indicates using two-step finetuning.

| Model | #Params | PPL |
|---|---|---|
| Shazeer et al. (2014) – Sparse Non-Negative | 33B | 52.9 |
| Chelba et al. (2013) – RNN-1024 + 9 Gram | 20B | 51.3 |
| Jozefowicz et al. (2016) – LSTM-2048-512 | 0.83B | 43.7 |
| Kuchaiev & Ginsburg (2017) – BIG G-LSTM-2 | - | 36.0 |
| Dauphin et al. (2016) – GCNN-14 bottleneck | - | 31.9 |
| Jozefowicz et al. (2016) – LSTM-8192-1024 | 1.8B | 30.6 |
| Jozefowicz et al. (2016) – LSTM-8192-1024 | 1.04B | 30.0 |
| Shazeer et al. (2017) – Low-Budget MoE | $\sim$5B | 34.1 |
| Shazeer et al. (2017) – High-Budget MoE | $\sim$5B | 28.0 |
| Ours – Transformer-XL | 0.46B | **23.5** |

Table 4: Comparison with state-of-the-art results on One Billion Word.

larization, Transformer-XL achieves a new SoTA result among models without two-step finetuning. Penn Treebank has only 1M training tokens, which implies that Transformer-XL can still generalize well even on small datasets.

## 4.2 ABLATION STUDY

We conduct an ablation study on WikiText-103 to examine the effects of two proposed techniques used in Transformer-XL: the recurrence mechanism and the new positional encoding scheme. The results are reported in Table 5. Among the compared encoding schemes, Shaw et al. (2018) is relative, while Vaswani et al. (2017) and Al-Rfou et al. (2018) are absolute. "Full" and "half" losses refer to applying a cross entropy loss to all or the recent half positions in the segment. We found that absolute encodings only work well with half losses because half losses exclude positions with very short attention lengths during training for better generalization. Table 5 shows that both the recurrence mechanism and our encoding scheme are necessary to achieve the best performance, as well as generalizing to longer attention sequences during evaluation time. Although the backpropagation length during training is only 128, with the two techniques the attention length can be increased to 640 at test time. In the standard setting with 151M parameters, the perplexity decreases as the attention length increases. This is consistent with our motivation and discussion in Section 3.

Since the recurrence mechanism costs additional memory usage, we also compare Transformer-XL with baselines under the same GPU memory constraints. As shown in Table 8 in Appendix A, despite using a shorter backpropagation length, Transformer-XL remains superior to the baselines.

## 4.3 RELATIVE EFFECTIVE CONTEXT LENGTH

Khandelwal et al. (2018) proposed a method to evaluate the *Effective Context Length* (ECL) of a sequence model. ECL is the longest length to which increasing the context span would lead to a gain

| Remark | Recurrence | Encoding | Loss | PPL init | PPL best | Attn Len |
|---|---|---|---|---|---|---|
| Transformer-XL (128M) | ✓ | Ours | Full | **27.02** | **26.77** | **500** |
| - | ✓ | Shaw et al. (2018) | Full | 27.94 | 27.94 | 256 |
| - | ✓ | Ours | Half | 28.69 | 28.33 | 460 |
| - | ✗ | Ours | Full | 29.59 | 29.02 | 260 |
| - | ✗ | Ours | Half | 30.10 | 30.10 | 120 |
| - | ✗ | Shaw et al. (2018) | Full | 29.75 | 29.75 | 120 |
| - | ✗ | Shaw et al. (2018) | Half | 30.50 | 30.50 | 120 |
| - | ✗ | Vaswani et al. (2017) | Half | 30.97 | 30.97 | 120 |
| Transformer (128M)[†] | ✗ | Al-Rfou et al. (2018) | Half | 31.16 | 31.16 | 120 |
| Transformer-XL (151M) | ✓ | Ours | Full | 23.43 | **23.09** | **640** |
|  |  |  |  |  | 23.16 | 450 |
|  |  |  |  |  | 23.35 | 300 |

Table 5: Ablation study on WikiText-103. For the first two blocks, we use a slightly smaller model (128M parameters). † indicates that the corresponding row is reduced to the same setting as the Transformer network in Al-Rfou et al. (2018), except that two auxiliary losses are not implemented in our experiments. "PPL init" refers to using the same length as training. "PPL best" indicates the perplexity obtained by using the optimal length. "Attn Len" is the shortest possible attention length during evaluation to achieve the corresponding result (PPL best). Increasing the attention length during evaluation improves performance only when our positional encoding is used. The "Transformer-XL (151M)" setting uses a standard parameter budget as previous work Merity et al. (2018), where we observe a similar effect when increasing the attention length during evaluation.

| Model | $r = 0.1$ | $r = 0.5$ | $r = 1.0$ |
|---|---|---|---|
| Transformer-XL 151M | **900** | **800** | **700** |
| QRNN | 500 | 400 | 300 |
| LSTM | 400 | 300 | 200 |
| Transformer-XL 128M | **700** | **600** | **500** |
| - use Shaw et al. (2018) encoding | 400 | 400 | 300 |
| - remove recurrence | 300 | 300 | 300 |
| Transformer | 128 | 128 | 128 |

Table 6: Relative effective context length (RECL) comparison. See text for the definition of RECL and $r$. The first three models and the last four models are compared as two *model groups* when we calculate RECL (RECL is computed on a model group rather than a single model). Each group has the same parameter budget.

more than a threshold. However, ECL ignores the fact that it is harder to get improvement when a model already achieves a lower perplexity using only a shorter context, and thus it is not suitable for fair comparison among multiple models. We instead propose a new metric called *Relative Effective Context Length* (RECL). RECL is defined on a model group instead of a single model, and the gain of a long context is measure by the relative improvement over the *best* short context model. As such, the model group shares the same baseline to enable fair comparison. RECL also has a parameter $r$, which means constraining the comparison on top-$r$ hard examples. See Appedix C for more details about RECL. As shown in Table 6, Transformer-XL manages to model dependency of 900 words long on average with $r = 0.1$. The RECL of Transformer-XL is 80% and 450% longer than recurrent networks and Transformer respectively. Both the recurrence mechanism and our positional encodings contribute to a longer RECL. This further substantiates our argument that Transformer-XL is able to model longer-term dependency.

## 4.4 EVALUATION SPEED

Finally, we compare the evaluation speed of the proposed model with the vanilla Transformer model Al-Rfou et al. (2018). As shown in Table 7, due to the state reuse scheme, Transformer-XL achieves an up to 1,874 times speedup during evaluation compared to the architecture in Al-Rfou et al. (2018).

| Attn Len | How much Al-Rfou et al. (2018) is slower than ours |
|---|---|
| 3,800 | 1,874x |
| 2,800 | 1,409x |
| 1,800 | 773x |
| 800 | 363x |

Table 7: Slowdown in terms of computational time during evaluation. Evaluation is based on per-token time on one GPU.

## 5 CONCLUSIONS

We propose a novel architecture, Transformer-XL, for language modeling with longer-term dependency. Our main technical contributions include introducing the notion of recurrence in a purely self-attentive model and deriving a novel positional encoding scheme. Transformer-XL is the first self-attention model that achieves substantially better results than RNNs on both character-level and word-level language modeling. Transformer-XL is also able to model longer-term dependency than RNNs and Transformer.

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

## A    ABLATION STUDY WITH MEMORY CONSTRAINTS

| Backprop Len | Recurrence | Encoding | Loss | pplx best | pplx init | Attn Len |
|:---:|:---:|:---:|:---:|:---:|:---:|:---:|
| 128 | ✓ | Ours | Full | **26.77** | **27.02** | **500** |
| 128 | ✓ | Ours | Partial | 28.33 | 28.69 | 460 |
| 176 | ✗ | Ours | Full | 27.98 | 28.43 | 400 |
| 172 | ✗ | Ours | Partial | 28.83 | 28.83 | 120 |

Table 8: Ablation study on WikiText-103 with the same GPU memory constraints.

Table 8 compares Transformer-XL with baseline under the same memory budget. Transformer-XL still outperforms the baseline even with a shorter backprop length.

## B    EFFICIENT COMPUTATION OF THE ATTENTION WITH RELATIVE POSITIONAL EMBEDDING

As we discussed in section 3.3, the naive way of computing the $\mathbf{W}_{k,R}\mathbf{R}_{i-j}$ for all pairs $(i, j)$ is subject to a quadratic cost. Here, we present a simple method with only a linear cost. Firstly, notice that the relative distance $i - j$ can only be integer from 0 to $M + L - 1$, where $M$ and $L$ are the memory length and segment length respectively. Hence, the rows of the matrix

$$\mathbf{Q} := \begin{bmatrix} \mathbf{R}_{M+L-1}^{\top} \\ \mathbf{R}_{M+L-2}^{\top} \\ \vdots \\ \mathbf{R}_{1}^{\top} \\ \mathbf{R}_{0}^{\top} \end{bmatrix} \mathbf{W}_{k,R}^{\top} = \begin{bmatrix} [\mathbf{W}_{k,R}\mathbf{R}_{M+L-1}]^{\top} \\ [\mathbf{W}_{k,R}\mathbf{R}_{M+L-2}]^{\top} \\ \vdots \\ [\mathbf{W}_{k,R}\mathbf{R}_{1}]^{\top} \\ [\mathbf{W}_{k,R}\mathbf{R}_{0}]^{\top} \end{bmatrix} \in \mathbb{R}^{(M+L) \times d}$$

consist of all possible vector outputs of $\mathbf{W}_{k,R}\mathbf{R}_{i-j}$ for any $(i, j)$. Note that we have defined $\mathbf{Q}$ in a reversed order, i.e., $\mathbf{Q}_k = \mathbf{W}_{k,R}\mathbf{R}_{M+L-1-k}$, to make further discussion easier.

Next, we collect the term $(b)$ for all possible $i, j$ into the following $L \times (M + L)$ matrix,

$$\mathbf{B} = \begin{bmatrix} q_0^{\top}\mathbf{W}_{k,R}\mathbf{R}_M & \cdots & q_0^{\top}\mathbf{W}_{k,R}\mathbf{R}_0 & 0 & \cdots & 0 \\ q_1^{\top}\mathbf{W}_{k,R}\mathbf{R}_{M+1} & \cdots & q_1^{\top}\mathbf{W}_{k,R}\mathbf{R}_1 & q_1^{\top}\mathbf{W}_{k,R}\mathbf{R}_0 & \cdots & 0 \\ \vdots & \vdots & \vdots & \vdots & \ddots & \vdots \\ q_{L-1}^{\top}\mathbf{W}_{k,R}\mathbf{R}_{M+L-1} & \cdots & q_{L-1}^{\top}\mathbf{W}_{k,R}\mathbf{R}_{M+L-1} & q_{L-1}^{\top}\mathbf{W}_{k,R}\mathbf{R}_{L-1} & \cdots & q_{L-1}^{\top}\mathbf{W}_{k,R}\mathbf{R}_0 \end{bmatrix}$$

$$= \begin{bmatrix} q_0^{\top}\mathbf{Q}_{L-1} & \cdots & q_0^{\top}\mathbf{Q}_{M+L-1} & 0 & \cdots & 0 \\ q_1^{\top}\mathbf{Q}_{L-2} & \cdots & q_1^{\top}\mathbf{Q}_{M+L-2} & q_1^{\top}\mathbf{Q}_{M+L-1} & \cdots & 0 \\ \vdots & \vdots & \ddots & \vdots & \ddots & \vdots \\ q_{L-1}^{\top}\mathbf{Q}_0 & \cdots & q_{L-1}^{\top}\mathbf{Q}_M & q_{L-1}^{\top}\mathbf{Q}_{M+1} & \cdots & q_{L-1}^{\top}\mathbf{Q}_{M+L-1} \end{bmatrix}$$

Then, we further define

$$\widetilde{\mathbf{B}} = \mathbf{q}\mathbf{Q}^{\top} = \begin{bmatrix} q_0^{\top}\mathbf{Q}_0 & \cdots & q_0^{\top}\mathbf{Q}_M & q_0^{\top}\mathbf{Q}_{M+1} & \cdots & q_0^{\top}\mathbf{Q}_{M+L-1} \\ q_1^{\top}\mathbf{Q}_0 & \cdots & q_1^{\top}\mathbf{Q}_M & q_1^{\top}\mathbf{Q}_{M+1} & \cdots & q_1^{\top}\mathbf{Q}_{M+L-1} \\ \vdots & \vdots & \ddots & \vdots & \ddots & \vdots \\ q_{L-1}^{\top}\mathbf{Q}_0 & \cdots & q_{L-1}^{\top}\mathbf{Q}_M & q_{L-1}^{\top}\mathbf{Q}_{M+1} & \cdots & q_{L-1}^{\top}\mathbf{Q}_{M+L-1} \end{bmatrix}.$$

Now, it is easy to see an immediate relationship between $\mathbf{B}$ and $\widetilde{\mathbf{B}}$, where the $i$-th row of $\mathbf{B}$ is simply a left-shifted version of $i$-th row of $\widetilde{\mathbf{B}}$. Hence, the computation of $\mathbf{B}$ only requires a matrix multiplication $\mathbf{q}\mathbf{Q}^{\top}$ to compute $\widetilde{\mathbf{B}}$ and then a set of left-shifts.

Similarly, we can collect all term $(d)$ for all possible $i, j$ into another $L \times (M + L)$ matrix $\mathbf{D}$,

$$\mathbf{D} = \begin{bmatrix} v^\top \mathbf{Q}_{L-1} & \cdots & v^\top \mathbf{Q}_{M+L-1} & 0 & \cdots & 0 \\ v^\top \mathbf{Q}_{L-2} & \cdots & v^\top \mathbf{Q}_{M+L-2} & v^\top \mathbf{Q}_{M+L-1} & \cdots & 0 \\ \vdots & \vdots & \ddots & \vdots & \ddots & \vdots \\ v^\top \mathbf{Q}_0 & \cdots & v^\top \mathbf{Q}_M & v^\top \mathbf{Q}_{M+1} & \cdots & v^\top \mathbf{Q}_{M+L-1} \end{bmatrix}.$$

Then, we can follow the same procedure to define

$$\widetilde{\mathbf{d}} = [\mathbf{Q}v]^\top = \begin{bmatrix} v^\top \mathbf{Q}_0 & \cdots & v^\top \mathbf{Q}_M & v^\top \mathbf{Q}_{M+1} & \cdots & v^\top \mathbf{Q}_{M+L-1} \end{bmatrix}.$$

Again, each row of $\mathbf{D}$ is simply a left-shift version of $\widetilde{\mathbf{d}}$. Hence, the main computation cost comes from the matrix-vector multiplication $\widetilde{\mathbf{d}} = [\mathbf{Q}v]^\top$, which is not expensive any more.

## C    DETAILS ABOUT RECL

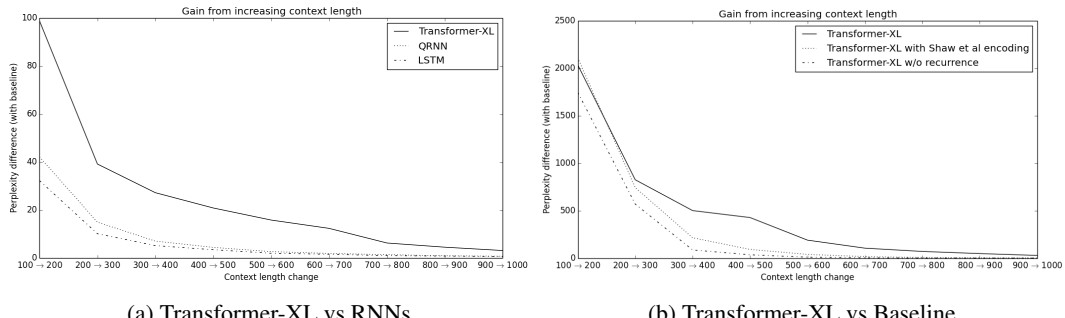

(a) Transformer-XL vs RNNs          (b) Transformer-XL vs Baseline

Figure 3: Visualizing unnormalized relative perplexity gains with $r = 0.1$.

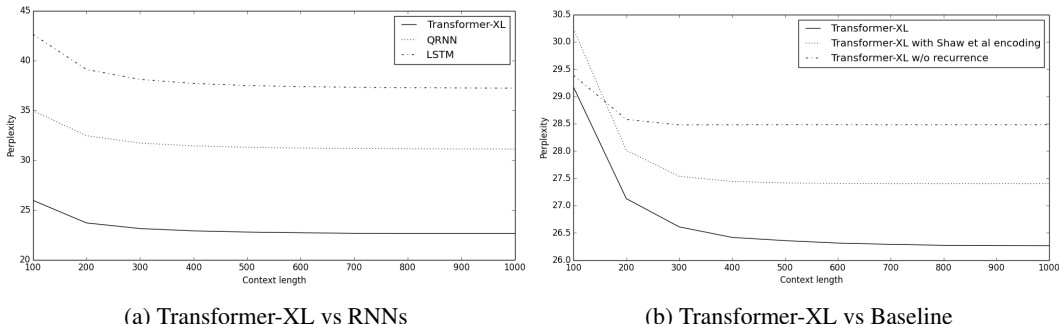

(a) Transformer-XL vs RNNs          (b) Transformer-XL vs Baseline

Figure 4: Perplexity vs context length.

In this section, we describe the details of the metric RECL. Let $\mathcal{M} = \{m_1, m_2, \cdots, m_N\}$ be a model group consisting of $N$ models. Let $l_i(c, t)$ denote the loss of model $m_i$ on the $t$-th token in the corpus with a context length $c$. Concretely, the loss can be written as

$$l_i(c, t) = -\log P_{m_i}(x_t | x_{t-1}, \cdots, x_{t-c})$$

where $P_{m_i}$ is the probability distribution given by model $m_i$, and $x_t$ is the $t$-th token in the corpus. Given a short context length $c$ and a long context length $c'$ such that $c' \geq c$, we can further define a baseline for each position $t$,

$$b(c, t) = \min_{i=1}^{N} l_i(c, t)$$

The *relative loss* of $m_i$ w.r.t. the model group $\mathcal{M}$ is written as

$$f_i(c, c') = \frac{1}{|\mathcal{T}|} \sum_{t \in \mathcal{T}} \min \left( b(c, t), l_i(c', t) \right)$$

The above equation uses the minimum loss of all models on the short length $c$ as a baseline, and only losses smaller than the baseline will be effectively counted towards the relative loss. This enables fair comparison between multiple models because all models with a long context length $c'$ need to improve over the same baseline. Sometimes we only care about those positions where the baseline performs poorly (which means short-term dependency with context length $c$ is not sufficient), so given a ratio parameter $r$, we define the set $\mathcal{T}$ is the above equation as

$$\mathcal{T} = \text{top-}r \text{ positions } t \text{ with largest } b(c, t)$$

The *relative gain* is subsequently defined as the relative perplexity reduction:

$$g_i(c, c') = \frac{\exp f_i(c, c) - \exp f_i(c, c')}{\exp f_i(c, c)}$$

Given a step size $\Delta$, we then use an algorithm to find the RECL by thresholding the relative gain:

1. Set initial short context length $c$, and long context length $c' = c + \Delta$
2. Compute $g_i(c, c')$. If $g_i(c, c') < 0.01$, return RECL $= c$. If $g_i(c, c') \geq 0.01$, set $c = c', c' = c + \Delta$ and go to step 1.

In Figure 3, we visualize the unnormalized relative perplexity gains $(\exp f_i(c, c) - \exp f_i(c, c'))$ with various pairs of $(c, c')$ when $r = 0.1$. It is clear that Transformer-XL has a longer RECL compared to RNNs and other baselines because the relative gains are substantially larger.

For reference, we plot the perplexities with varying context lengths in Figure 4. The y-axis denotes the "normal" perplexity (not calibrated by baselines).

## D   ATTENTION VISUALIZATION

In this section, we provide some visualization of the attention learned by the SoTA model on the WikiText-103 validation set. Recall that, this model has 16 10-head transformer layers and relies on a memory of length 640.

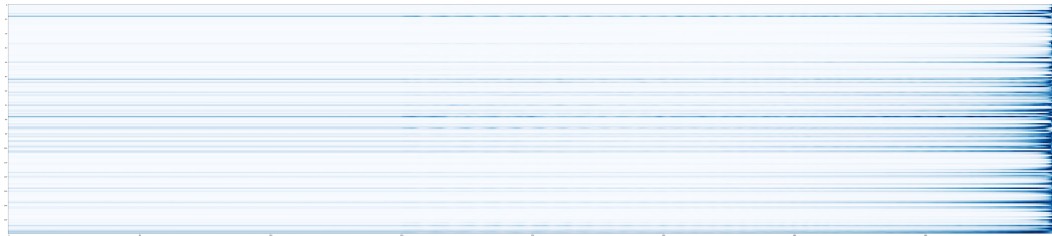

Figure 5: Average attention over the previous 640 tokens, where each row corresponds to a attention head and each column corresponds to a relative location. There are totally 160 attention heads, and every 10 heads come from a single layer. Darker colors indicate higher values.

The first visualization aims at revealing the overall trend of where the model is attending. Specifically, for each attention head of each layer, we average the attention distributions of all tokens in the validation set. This is shown in Fig. 5. As we can see, the overall trend is to focus more on the nearby tokens than the faraway ones. However, it is also very clear that some attention heads have a wider attention distribution over the entire memory span, notably the head 8 from layer 1, head 78 from layer 8, and the head 158 from layer 16.

Since we are focused on learning long-range dependency, we are especially interested in these heads with a wider attention span. Thus, in the second set of visualization, we pick the three notable heads mentioned above, and visualize their attention behavior for a randomly chosen position, as shown in Fig. 6. Here, we see three different patterns of wider attention:

- For the head 8 in the 1st layer, we see an almost uniform attention over the entire memory span. This is quite intuitive, as lower-level layers needs to screen the entire memory span to decide where to focus for higher-level layers

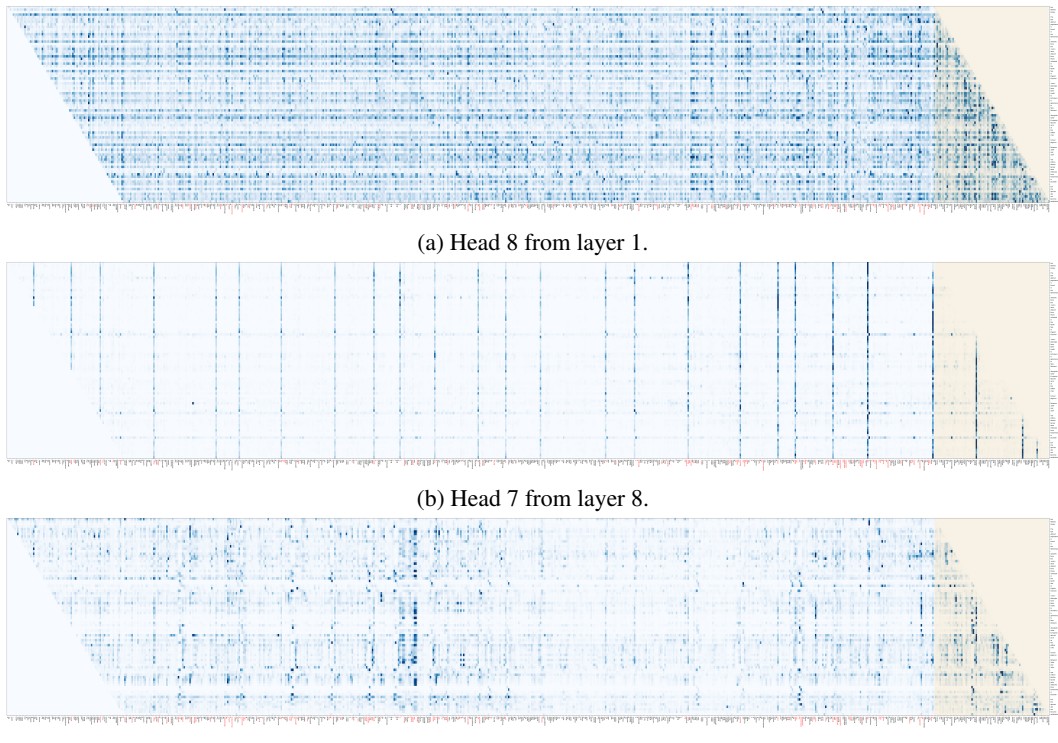

(a) Head 8 from layer 1.

(b) Head 7 from layer 8.

(c) Head 158 from layer 16.

Figure 6: Visualization of the three heads with a wide attention range. Each row corresponds to a target location/token and each column corresponds to a context location/token. Tokens in the memory that have top 20% attention values are highlighted in red.

- For the head 78 in the 8th layer (a middle-level layer), we see a very sparse attention pattern scattered in all ranges of the memory. Again, this well fits our intuition that as information accumulates, the network may focus on some particular position with special interests.

- For the head 158 in the 16th layer (i.e. the last layer), each target location (corresponding to each row) has its own distinct sparse focus, differing from head 78 where target locations largely share the same attentive location in memory. Meanwhile, the pattern is also different from the case of head 8, where a few locations are clearly attended more than others.

Finally, as we have discussed in section 3.3, the attention score can be decomposed into four intuitive terms. Here, we want to further investigate how these four terms contribute to the overall attention trend in Fig. 5. Since the term $(c)$ represents the global content bias, i.e., the prior importance of each word regardless of the context, we will leave it out and focus on the terms $(a)$, $(b)$ and $(d)$. So, for each term, we take the Softmax w.r.t. the memory span and average the resulted distribution of all tokens in the validation set. The results are visualized in Fig. 7:

- Since term $(a)$ is fully content-based addressing, when averaging over all target words, the result is essentially uniform over the entire context, except for a few very close words, which are likely to be semantically similar to the target word.

- The overall trend of term $(b)$ highly resembles that of the entire attention distribution in Fig. 5. It suggests that the global trend of focusing on the nearby context is largely contributed by this content-dependent positional bias.

- The overall trend of term $(d)$ is also focusing more on nearby words. However, compared to the trend of term $(b)$, it is clearly flatter and biases towards a longer context.

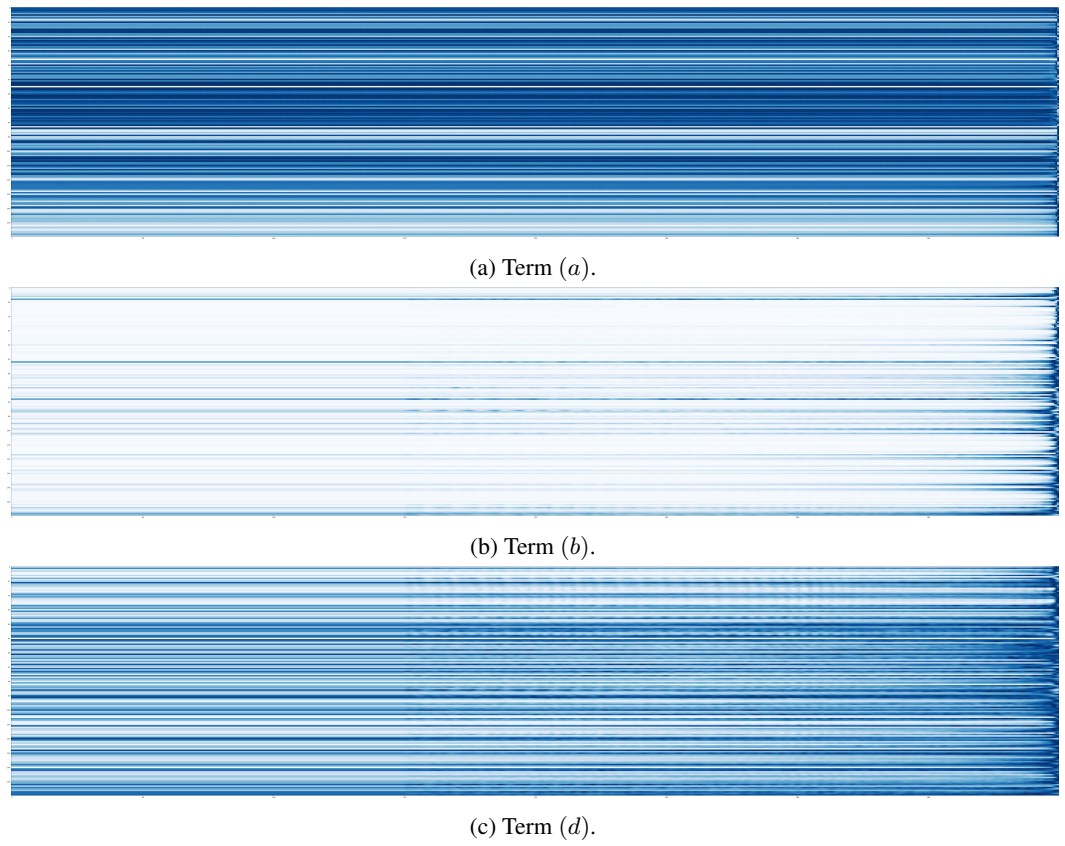

(a) Term $(a)$.

(b) Term $(b)$.

(c) Term $(d)$.

Figure 7: Visualization of the three terms in computing the attention score. Each row corresponds to a attention head and each column corresponds to a relative location.

