# OpenReview forum: "Transformer-XL: Language Modeling with Longer-Term Dependency"
_ICLR.cc/2019/Conference_

### Official Review · AnonReviewer1 · 2018-10-13
**Marginal innovation**

**Rating:** 4
**Confidence:** 4

**Review:**

This paper puts forward a new schema for language modeling, especially for relationship between two parts far apart.

The experimental results on WikiText-103 are good, improving the STOA PPL by 9.0. On the other three datasets, however, there's little or no gain. The speed comparison should be carried out over more LM models, as Al-Rfou is not the fastest.

The writing is not very clear, especially around equations.

Overall the contribution of this paper is marginally incremental:
1. The major proposed idea is just to add one no-grad previous segment into the prediction for next segment. This is similar to Residual network idea but more simplified.
2. Using relative positional encoding is not a new idea, e.g. https://arxiv.org/pdf/1803.02155.pdf.
3. Reusing previous level/segment computation with gradient fixed is also not a big innovation.

typo:
1. end of page 3, and "W." denotes".
2. The speed experiment should be put in the main text.

---

> ### Author Response · Authors · 2018-11-27
> **response**
>
> Thanks for your valuable comments!
>
> [Speed Comparison]
> As shown in our paper, Transformer is the state-of-the-art model on language modeling, and Al Rfou et al was the previous SoTA of Transformer language models. The main argument of our results on computational time is that Transformer-XL substantially improves the speed while getting even better results. It is less interesting to obtain speedup over a poorly performing model. On the other hand, as our speedup techniques specifically target Transformers, we believe Al Rfou et al is the most appropriate baseline to test the effects of our proposed methods.
>
> Please see our comments above regarding the significance and novelty of our contributions.

---

> ### Author Response · Authors · 2018-12-09
> **update**
>
> Dear reviewer, we believe we have addressed your concerns in the rebuttal (see the General Response above and the comments below). Especially, we have further improved over state-of-the-art results ever since. Do you have an updated assessment or other concerns of our paper? Thank you!

---

### Official Review · AnonReviewer3 · 2018-11-03
**Using Transformer as a RNN cell applied to equal-length segments, good experimental results, but need to cover standard benchmarks and use SOTA decoding techniques for comparison.**

**Rating:** 6
**Confidence:** 4

**Review:**

This paper proposes a Transformer based RNN structure "Transformer-XL" to capture long-range contextual relations and targets on language model task. The idea is straightforward: it splits the input sequence into equal and fixed length segments, and recurrently apply the Transformer over the sequence of segments, in which the hidden states for the previous segment are treated as a memory to attend for the next segment.

This paper is well-organized and well-written, and easy to follow. The empirical results also demonstrate the proposed model can achieve SoTA performance on several word- and character-based language model benchmarks.


Pros:

1. The model is designed based on a careful engineering: 1) taking into account the history hidden states for long-term dependency modeling and 2) alignment scores calculated from multiple perspectives for relative position modeling and global significance capturing. In addition, in contrast to the previous Transformer-based language model, benefiting from the recurrent architecture, both training and decoding can be accelerated.
2. The experimental results show that the proposed Transformer-XL can surpass the baseline model and achieve new state-of-the-art perplexity or bpc on word- or char-based language model task. And, based on the proposed new metric, RECL, the analysis for context length modeling verifies the proposed model can make the best of long-range dependencies.


Cons:

1. The proposed model is ad-hoc and is only compatible with language model task. Is it possible to extend the proposed model to more general and practical tasks (e.g., seq2seq tasks)?
2. The absence of a popular language model benchmark, WikiText-2, which has been evaluated in most previous papers.
3. It is notable that there are no ubiquitous decoding techniques for the language used in both the proposed model and baselines, such as dynamical evaluation and continuous cache pointer. However, these techniques are essential for the RNN-LM baselines to achieve state-of-the-art performance, and has been standardly used in most previous works. Therefore, the comparison seems unfair.

Minor comments: In Figure 1 and 2, it is better to include a legend explaining the meaning of different colors for different nodes.

---

> ### Author Response · Authors · 2018-11-27
> **response**
>
> Thanks for your valuable comments!
>
> [WT2]
> WT2 shares the same test set as WT103, and the only difference is that WT103 has more training data. Since language modeling has almost unlimited training data in nature, we believe it brings less benefit to compare models on more small-scale datasets as we already have results on Penn Treebank which is also a small dataset.
>
> [Test-time evaluation techniques]
> In Table 1, we show that our method without any test-time evaluation techniques is still 21+ points better than Grave et al which employs test-time continuous cache on WT103. On enwiki8, mLSTM + dynamic eval [1] achieves a BPC of 1.08, which is still 0.09 worse than Transformer-XL without dynamic evaluation. On One Billion Word, the best previous result did not use test-time evaluation techniques. The only exception is Penn Treebank, where we exclude results with test-time techniques to focus on comparing different architectures. This is fair comparison because all considered models do not use test-time techniques. Moreover, according to previous results, test-time evaluation techniques bring consistent improvement to different architectures (Yang et al 2017, Merity et al 2017).
>
> Please see our comments above regarding the importance of language modeling on its own.
>
> [1] Ben Krause, Emmanuel Kahembwe, Iain Murray, and Steve Renals. Dynamic evaluation of
> neural sequence models.

---

> ### Author Response · Authors · 2018-12-09
> **update**
>
> Dear reviewer, we believe we have addressed your concerns in the rebuttal (see the General Response above and the comments below). Especially, we have further improved over state-of-the-art results ever since. Do you have an updated assessment or other concerns of our paper? Thank you!

---

### Official Review · AnonReviewer2 · 2018-11-05
**This paper proposes a variant of transformer to train language model**

**Rating:** 6
**Confidence:** 4

**Review:**

This paper proposes a variant of transformer to train language model, it uses two modifications, one is the segment level recurrence with state reuse, the other is relative positional encoding, which significantly enhances the power to model long range dependency. Extensive experiments in terms of perplexity results are reported, specially on WikiText-103 corpus, significant perplexity reduction has been achieved.

Perplexity is not a gold standard for language model, the authors are encouraged to report experimental results on real world applications such as word rate reduction ASR on BLEU score improvement machine translation.

Ciprian Chelba and Frederick Jelinek, Structured language modeling. Computer Speech and Language (2000) 14, 283–332.

Peng Xu, Frederick Jelinek: Random forests and the data sparseness problem in language modeling. Computer Speech & Language 21(1): 105-152 (2007).

---

> ### Author Response · Authors · 2018-11-27
> **response**
>
> Thanks for your valuable comments!
>
> As far as we know, almost all language models were evaluated by perplexity in previous work.
>
> Please see our comments above regarding the importance of language modeling on its own.

---

> ### Author Response · Authors · 2018-12-09
> **update**
>
> Dear reviewer, we believe we have addressed your concerns in the rebuttal (see the General Response above and the comments below). Especially, we have further improved over state-of-the-art results ever since. Do you have an updated assessment or other concerns of our paper? Thank you!

---

### Author Response · Authors · 2018-11-27
**General response to the reviewers and AC**


[Latest results and significance]
We have experimented with increasing the model sizes so as to match the previous work for fair comparison. As a result, we have advanced the SoTA performance from 1.06 to 0.99 in bpc on enwiki8, from 33.0 to 18.9 in perplexity on WT103, and from 28.0 to 23.5 in perplexity on One Billion Word. Note that our result on enwiki8 is the first result below 1.0 bpc on widely-studied char-level LM benchmarks. We believe the improvement is significant compared to any previous results and we also believe this substantiates the significance of the proposed methods. Changes have been made accordingly in the paper.


[Why language modeling]
Although we believe our technique will be useful where long-term dependency is involved, with applications like paragraph-level machine translation, summarization, multi-paragraph question answering, text generation, etc, we would also like to emphasize that language modeling itself is important.
--- Firstly, language modeling has been an independent research direction in natural language processing (NLP) for decades [1-5]. Even when we restrict our attention to neural language models in the last two years, there has been a significant amount of work focused solely on this topic [6-18] in venues like ICLR, ICML, and NeurIPS.
--- Secondly, language modeling is an important unsupervised pretraining objective. The biggest advance in NLP recently originated from training large-scale language models for unsupervised feature learning [19,20].

[1] Chen, S. F., & Goodman, J. (1996). An empirical study of smoothing techniques for language modeling
[2] Manning, C. D., & Schütze, H. (1999). Foundations of statistical natural language processing
[3] Bengio, Y., et. al. (2003). A neural probabilistic language model
[4] Mikolov, T. et al.  (2010) Recurrent neural network based language model
[5] Zaremba, W., et. al. (2014). Recurrent neural network regularization
[6] Jozefowicz, R., et. al. (2016). Exploring the limits of language modeling
[7] Grave, E., et. al. (2016). Improving neural language models with a continuous cache
[8] Press, O., & Wolf, L. (2016). Using the output embedding to improve language models
[9] Krause, B., et. al. (2016). Multiplicative LSTM for sequence modeling
[10] Merity, S., et. al. (2016). Pointer sentinel mixture models
[11] Dauphin, Y. N., et. al. (2017). Language modeling with gated convolutional networks
[12] Merity, S., et. al. (2017). Regularizing and optimizing LSTM language models
[13] Melis, G., et. al. (2017). On the state of the art of evaluation in neural language models.
[14] Yang, Z., et. al. (2017). Breaking the softmax bottleneck
[15] Merity, S., et. al. (2018). An Analysis of Neural Language Modeling at Multiple Scales
[16] Rae, J. W., et. al. (2018). Fast Parametric Learning with Activation Memorization
[17] Kanai, S., et. al. (2018). Sigsoftmax: Reanalysis of the Softmax Bottleneck
[18] Al-Rfou, R., et. al. (2018). Character-level language modeling with deeper self-attention.
[19] Peters, M., et. al. (2017) Deep contextualized word representations
[20] Devlin, J., et. al. (2018) BERT: Pre-training of Deep Bidirectional Transformers for Language Understanding


[Our contributions and novelty]
We believe Transformer-XL addresses an important problem. The key question we answer in this work is how to enable self-attention, an architecture which has a potential optimization advantage in learning long-term dependency, to really capture a longer context beyond a fixed length.

Our main contribution is to propose a complete set of techniques that jointly enable recurrency in self-attention, rather than a set of unrelated, individual techniques.
--- As described in Section 3 and shown in Table 5, state reuse is not even possible without relative positional encodings, because the absolute positions in the current segment are not the same as in the next segment.
---Similarly, relative positional encodings alone do not improve the ability to model long-term dependency.

Although our positional encodings share somewhat similar formulation to previous work such as Shaw et al, the motivation is completely different, and it is non-trivial to figure out such a combination of techniques for modeling long-term dependency with the self-attention architecture.

---

### Public Comment · ~Anirudh_Goyal1 · 2018-11-28
**Interesting paper**

Hello,

I believe this paper is addressing an interesting problem i.e enabling self attention to scale. And I enjoyed reading this paper! And results of this paper are pretty interesting too! :-)

Few points:

1. Generally while evaluating long term dependencies, I'm a bit skeptical about evaluating "bpc" or ppl. What has worked well for me in the past is to evaluate on longer sequences then it was trained for. As RNN's are generally trained using one step ahead prediction, so evaluating for longer sequences generally pose a more difficult problem. I personally always use this metric and only mostly use bpc for the sake of submitting papers. So, if authors have pre-trained model, I actually encourage them to use them and report  (same) metric on longer sequences, as compared to the respective baselines. I also think, this might make the paper stronger, and also, it may increase the chances of paper getting accepted. :-) Again, good work.


2. I would also like to point, that we had a paper in which we propose to enable self attention to scale, "Sparse Attentive Backtracking" (NIPS'18/NeurIPS'18), our motivation was  very different. https://arxiv.org/abs/1809.03702. It would be interesting if the authors can reference/cite this.

---

> ### Author Response · Authors · 2018-11-30
> **Response**
>
> Thanks for the comment.
>
> 1. We don’t fully understand the suggestion. One interpretation is to evaluate the RNN and the proposed model on longer sequences than that used in training. In this case, since truncated BPTT is used in the training, one can always pass the last-step hidden state from the previous segment to the next segment as the initial state of the RNN. Hence, the RNN is actually evaluated on the entire text sequence.
>
> 2. Thanks for pointing out this related work. We will check it and address the relationship properly in a later version.

---

### Public Comment · ~Noam_Shazeer1 · 2018-12-17
**Requesting details for billion-word-lm model hyperparameters**

Very impressive results!  For the billion-word benchmark, you are getting better perplexity numbers (23.5) than we have for models of comparable size (see https://arxiv.org/pdf/1811.02084.pdf).  Since, as you mention, context length is not an issue for this dataset, I would like to know what you are doing better so that we can improve our own results.  In particular, what are your settings for the following for the PPL=23.5 model:

Hyperparameters as defined in https://arxiv.org/pdf/1706.03762.pdf:
  1. Number of layers (n)
  2. Dimensionality of embedding matrices, layer inputs/outputs (d_model)
  3. Feed-forward hidden size (d_ff)
  4. Number of attention heads (h)
  5. key/value dimensionality (d_k), (d_v)
  7. Dropout rate
In addition:
  6. Did you use the original ~800K-word vocabulary or a character-level or word-piece-level encoding scheme
  7. Was your setup based on the tensor2tensor library or other open-source implementation?
  8. Dropout rates
  9. Number of training epochs

Thank you in advance for the clarification.

---

> ### Author Response · Authors · 2018-12-17
> **response**
>
> Thank you for your questions. We will publish our code along with our hyper-parameters on all the datasets very soon!

---

### Author Response · Authors · 2019-01-10
**[DEPRECATED] This version is outdated**

We have released a new version of this paper on arxiv https://arxiv.org/abs/1901.02860
along with code, pretrained models, hyperparameters, as well as new (even better) results.

Please refer to our arxiv version in your future work.

---

### Public Comment · ~Rajarshi_Das1 · 2019-01-20
**Variable "m" in equation (end of page 5)**

This paper makes great contributions and like many, I was sad to see it get rejected.

I was looking at the equations closely and the final equations describing the model has a variable "m" which hasn't been defined before. Specifically, I am referring to the "m" within the stop-gradient operator in the equation below.

$\tilde{h}_{\tau}^{n - 1} = [SG(m_{\tau}^{n-1}) \cdot h_{tau}^{n-1}]$

This set of equations does not have a direct dependence on $h_{\tau - 1}^{n-1}$ (the previous segment), so I am guessing "m" is capturing it somehow and it is not very clear presently.

Thank you in advance for the clarification.

---

> ### Author Response · Authors · 2019-01-20
> **response**
>
> Hi Rajarshi, thanks a lot for your comments. For the $m$, we define it in the paragraph right before section 3.3. It refers to the "memory", which can contain $h_{\tau - 1}^{n-1}$ or additionally more faraway segments like $h_{\tau - 2}^{n-1}$.

---

### Meta-Review · Area_Chair1 · 2018-12-14
**reject**

**Confidence:** 4
**Recommendation:** Reject

**Metareview:**

despite the (significant) improvement in language modelling, it has always been a thorny issue whether better language models (at this level) lead to better performance in the downstream task or whether such a technique could be used to build a better conditional language model which often focuses on the aspect of generation. in this context, the reviewers found it difficult to see the merit of the proposed approach, as the technique itself may be considered a rather trivial application of earlier approaches such as truncated backprop. it would be good to apply this technique to e.g. document-level generation and see if the proposed approach can strike an amazing balance between computational efficiency and generation performance.

---

> ### Author Response · Authors · 2018-12-26
> **response**
>
>
> === About the technical contribution ===
>
> Firstly, as we have explained in the rebuttal and paper, trivially applying truncated BPTT to Transformer will NOT work due to the temporal confusion caused by using the same absolute positional encoding on two consecutive segments. In this work, we identify that it is the temporal confusion problem which prevents the reuse of historical hidden states. More importantly, we figure out the confusion could be resolved by only injecting relative positional information. This process of identifying, analyzing and solving the problem is a non-trivial scientific process, as no other previous or contemporary work targeting at using self-attention for language modeling has provided such a solution despite the fact that everyone working in LM is familiar with truncated BPTT.
>
> Additionally, to facilitate the learning of the recurrence mechanism, we also propose a more generalizable relative positional encoding and establish its non-trivial performance advantage in ablation.
>
> Hence, we respectfully disagree with the argument that the proposed approach is “a rather trivial application of earlier approaches such as truncated backprop.”
>
> === About the value of enabling recurrence for self-attention in the context of LM ===
>
> We think the question can be broken into three sub-questions with different levels:
> (1) Is a better language model by itself important or not?
> (2) What is the application value of a better self-attention LM that can utilize recurrence?
> (3) Is it useful to create recurrence in self-attention in general, e.g., beyond the text domain?
>
> The question (1) essentially asks whether we need a better density estimator for text. The answer to this question can be rather subjective and differ from person to person. That said, as one of the most fundamental statistical questions, density estimation should have its scientific values.
>
> For question (2), it is not difficult to come up with a list of potential applications. Firstly, many document-level problems could benefit from the proposed model, such as the document-level summarization, translation, seqeuntial labeling, and reading comprehension. Note that these tasks don’t have to be restricted to text generation. Secondly, besides serving as an architecture for downstream tasks, language models can also be used to perform “unsupervised feature learning” as demonstrated by recent advancement in NLP [1,2]. Hence, given a language model that can better capture the contextual information, it is very likely the hidden representations within the language model are also superior.
>
> Finally, question (3) is concerned with whether the techniques proposed in this work can be applied to other domains other than language. On this matter, we believe there exists a common desire of capturing longer-term dependency in sequence modeling. For example, in the speech domain, the raw data often has a sample rate of 16K Hz, which means that each second of speech data is a sequence of 16K steps. Similarly, in the domain of time seriers analysis (e.g. sensor data), the sequence length can also be very long.
>
> In summary, we believe language modeling is a reasonable testbed of model and algorithm development for NLP and more broadly sequence modeling.
>
> --------------------------------
> [1] Peters, M., et. al. (2017) Deep contextualized word representations
> [2] Devlin, J., et. al. (2018) BERT: Pre-training of Deep Bidirectional Transformers for Language Understanding